# Higher Serum Total Cholesterol to High-Density Lipoprotein Cholesterol Ratio Is Associated with Increased Mortality among Incident Peritoneal Dialysis Patients

**DOI:** 10.3390/nu14010144

**Published:** 2021-12-29

**Authors:** Hee-Won Noh, Yena Jeon, Ji-Hye Kim, Ga-Young Lee, Soo-Jee Jeon, Kyu-Yeun Kim, Jeong-Hoon Lim, Hee-Yeon Jung, Ji-Young Choi, Sun-Hee Park, Chan-Duck Kim, Yong-Lim Kim, Jang-Hee Cho

**Affiliations:** 1Division of Nephrology, Department of Internal Medicine, School of Medicine, Kyungpook National University, Daegu 41944, Korea; hwn1104@gmail.com (H.-W.N.); js1725@hanmail.net (J.-H.K.); aiyaiyai@naver.com (G.-Y.L.); hot716@naver.com (S.-J.J.); kgy8244@naver.com (K.-Y.K.); hunph84@naver.com (J.-H.L.); hy-jung@knu.ac.kr (H.-Y.J.); jyss1002@hanmail.net (J.-Y.C.); sh-park@knu.ac.kr (S.-H.P.); drchkim@knu.ac.kr (C.-D.K.); 2Department of Statistics, Kyungpook National University, Daegu 41566, Korea; yeahnah@naver.com; 3Cell and Matrix Research Institute, Kyungpook National University, Daegu 41944, Korea

**Keywords:** total cholesterol, high-density lipoprotein cholesterol, mortality, peritoneal dialysis

## Abstract

This study evaluated the association of the serum total cholesterol to high-density lipoprotein cholesterol ratio (TC/HDL-C) with mortality in incident peritoneal dialysis (PD) patients. We performed a multi-center, prospective cohort study of 630 incident PD patients from 2008 to 2015 in Korea. Participants were stratified into quintiles according to baseline TC, HDL-C, LDL-C and TC/HDL-C. The association between mortality and each lipid profile was evaluated using multivariate Cox regression analysis. During a median follow-up period of 70.3 ± 25.2 months, 185 deaths were recorded. The highest TC/HDL-C group had the highest body mass index, percentage of diabetes and serum albumin level. Multivariate analysis demonstrated that the highest quintile of TC/HDL-C was associated with increased risk of all-cause mortality (hazard ratio 1.69, 95% confidence interval 1.04–2.76; *p* = 0.036), whereas TC, HDL-C and LDL-C were not associated with mortality. Linear regression analysis showed a positive correlation between TC/HDL-C and body mass index. Increased serum TC/HDL-C was an independent risk factor for mortality in the subgroup of old age, female, cardiovascular disease and low HDL-C. The single lipid marker of TC or HDL-C was not able to predict mortality in PD patients. However, increased serum TC/HDL-C was independently associated with all-cause mortality in PD patients.

## 1. Introduction

The number of patients with end-stage renal disease (ESRD) requiring maintenance dialysis is increasing worldwide [1]. Although gradual improvements in dialysis treatment have been achieved, patients with ESRD have higher mortality rates when compared to patients with malignancy [1]. The high prevalence of cardiovascular disease (CVD) is considered a major contributor to early mortality in ESRD patients [1,2].

Hypercholesterolemia is a conventional risk factor of CVD in the general population. Both elevated serum total cholesterol (TC) and reduced high-density lipoprotein cholesterol (HDL-C) are associated with CVD. Recent studies show non-traditional lipid profiles, such as the TC to HDL-C ratio (TC/HDL-C), as independent lipoprotein predictors of the development of CVD. High TC/HDL-C is associated with increased risk of left ventricular hypertrophy, coronary heart disease, and ischemic stroke independently of plasma low-density lipoprotein cholesterol (LDL-C) levels [3,4,5,6].

A number of studies reported that a high TC/HDL-C ratio is associated with all-cause and cardiovascular mortality in incident hemodialysis (HD) patients. In contrast to the general population, a cohort study found that a low TC/HDL-C ratio was associated with higher mortality in HD patients [7]. A further study found that decreases in non-HDL-C and non-HDL-C/HDL-C contributed to increased all-cause and cardiovascular mortality in patients undergoing incident hemodialysis [8].

However, there is a lack of studies evaluating the association between non-traditional lipid profiles and CVD risk in peritoneal dialysis (PD) patients. PD patients typically have elevated levels of serum TC and LDL-C, whereas HD patients have normal or low levels of serum TC and LDL-C [9]. As PD patients have different lipid profiles compared to HD patients, non-traditional lipid profiles may have a different association with mortality in PD patients. In addition, a strategy to reduce LDL-C demonstrated no beneficial effect on CVD in dialysis patients [10]. Therefore, it may not be appropriate to evaluate the risk of CVD or mortality using only traditional lipid profiles in PD patients.

In the present study, we investigated the relationship between TC/HDL-C and mortality among incident PD patients. We evaluated the predictive value of TC/HDL-C in detecting the risk of CVD compared with individual traditional lipid profiles alone.

## 2. Materials and Methods

### 2.1. Study Population and Data Sources

A total of 630 incident PD patients were enrolled from 2008 to 2015 in a multi-center prospective cohort study conducted in Korea. All enrolled patients were aged 19 years or older who had received at least three consecutive months of PD therapy. Baseline serum TC and HDL-C levels were recorded. Patients who died within three months of dialysis initiation or without recorded baseline TC and HDL-C levels were excluded from the analysis.

This study was conducted in compliance with the ethical principles of the Helsinki declaration and approved by the institutional review board of the Kyungpook National University hospital. Written informed consent was obtained from all participants.

Patients were followed up until death, conversion to HD, or kidney transplantation. Baseline data were collected at the beginning of PD, including age at initiation of dialysis, sex, body mass index, primary renal disease, comorbidity at the initiation of dialysis (congestive heart failure, coronary artery disease, peripheral vascular disease, arrhythmia, cerebrovascular disease, chronic lung disease, peptic ulcer disease, chronic liver disease, connective tissue disease, and malignancy) and lab data at the initiation of dialysis (hemoglobin, blood urea nitrogen [BUN], creatinine, albumin, calcium, phosphorus, urine volume). Modified Charlson comorbidity index values were calculated [11].

### 2.2. Primary and Secondary Outcomes

Primary and secondary outcomes were all-cause and CVD mortality. CVD included coronary artery disease (angioplasty, coronary artery bypass grafting, and unstable angina), congestive heart failure (typical symptoms combined with systolic or diastolic dysfunction by echocardiography), cerebrovascular disease (ischemia or hemorrhagic infarct confirmed both by typical clinical symptoms and imaging study), and peripheral artery disease (revascularization or amputation).

### 2.3. Statistical Analyses

A preliminary analysis revealed that TC/HDL-C had a U-shape relationship with mortality rather than a linear relationship. Thus, participants were stratified into quintiles of TC/HDL-C (Q1 ≤ 3.11; 3.11 < Q2 ≤ 3.85; 3.85 < Q3 ≤ 4.6; 4.6 < Q4 ≤ 5.6; 5.6 < Q5). For a comparison of continuous and categorical variables among multiple groups, ANOVA and the Chi-square test, respectively, were used. The cumulative survival rates in each group were estimated by the Kaplan–Meier method and compared by using the log-rank test. Linear regression analyses were performed to identify the relationship between TC/HDL-C and body mass index (BMI), diabetes mellitus (DM), peptic ulcer disease, chronic liver disease, BUN, and albumin.

Multivariate Cox regression analyses were performed to evaluate the association between all-cause mortality and each lipid profile. The following 4 models were examined on the basis of multivariate adjustment: model 1, univariate hazard ratio of TC/HDL-C; model 2, model 1 plus age, sex, and BMI; model 3, model 2 plus hemoglobin, BUN, creatinine, albumin, calcium, and phosphorus; and model 4, model 3 plus 24 h urine volume. Statistical significance was defined as *p* < 0.05. Statistical analyses were performed using SPSS.

## 3. Results

### 3.1. Patient Baseline Characteristics

Baseline demographics, clinical, and laboratory characteristics according to baseline serum TC/HDL-C quintiles are summarized in Table 1. The mean age of patients at the initiation of dialysis was 51.5 ± 12.7 years; 59.4% were male, and the mean BMI was 23.1 ± 3.3 kg/m^2^. The mean ± standard deviation of TC and HDL-C were 168.4 ± 49.1 mg/dL and 41.0 ± 15.6 mg/dL, respectively. There were significant differences in BMI, prevalence of diabetes, prevalence of peptic ulcer disease, prevalence of liver disease, BUN and serum albumin levels among the TC/HDL-C quintiles. A linear regression analysis demonstrated that patients with elevated TC/HDL-C tended to have a higher BMI and serum BUN levels.

### 3.2. All-Cause and Cardiovascular Mortality

The mean follow-up duration of PD patients was 70.3 ± 25.2 months. During the follow-up period, 185 patients died. Among them, 73 (39.5%) were from cardiovascular disease and 36 (19.5%) from infectious disease.

Kaplan–Meier survival estimates of all-cause mortality with different levels of TC/HDL-C are shown in Figure 1. All-cause mortality in the combined first-to-fourth TC/HDL-C quintiles at 25, 50, 75, and 100 months was 4.95%, 17.4%, 25.8%, and 33.0%, respectively. On the other hand, all-cause mortality at 25, 50, 75, and 100 months was 8%, 20.9%, 34.7%, and 45.9%, respectively, in the fifth quintile group. There was a trend toward lower survival in the highest TC/HDL-C quintile compared to other groups (*p* = 0.052).

The multivariate Cox regression analysis found a decrease in patient survival in the highest TC/HDL-C group in PD patients (Table 2). The highest quintile group for baseline TC/HDL-C was associated with the highest all-cause mortality at all four levels of adjustment.

The reference group was quintile 3 (3.85 < TC/HDL-C ≤ 4.6). The hazard ratio (HR) of all-cause mortality in the fifth quintile was 1.74 (95% CI (Confidence interval), 1.09–2.76; *p* = 0.020) for model 1; 1.65 (95% CI, 1.04–2.64; *p* = 0.035) for model 2; 1.70 (95% CI, 1.04–2.76; *p* = 0.034) for model 3; and 1.69 (95% CI, 1.04–2.76; *p* = 0.036) for model 4. When the quintile 1 was considered as a reference group, there was no statistically significant difference from other groups (Appendix A). This result suggests that TC/HDL-C has clinical significance only when TC/HDL-C is very high.

A linear regression analysis showed a positive correlation between TC/HDL-C and BMI (Table 3). This result indicates that TC/HDL-C may be a risk factor for death in PD patients when there is a relationship between higher BMI and mortality.

We found no significant association between TC/HDL-C and CV mortality. Additionally, we analyzed a relationship between all-cause mortality and other lipid profiles. We found no significant differences in mortality between TC, HDL-C, and LDL-C quintiles at all four levels of adjustment (Table 4, Table 5 and Appendix A).

### 3.3. Subgroup Analyses

Subgroup analyses were performed in patients categorized according to age of 50 years, sex, presence of DM, 24 h urine volume of 100 mL/day, presence of CVD and BMI of 23 kg/m^2^. The negative association between high TC/HDL-C and survival was significant for patients with older age (≥50 years) and females. Higher TC/HDL-C was associated with the all-cause mortality rate in patients with CVD (Figure 2).

We also divided patients dichotomously according to median values of each serum lipid level as follows: TC < 162.5 mg/dL and ≥162.5 mg/dL, LDL-C < 93.8 mg/dL and ≥93.8 mg/dL, HDL-C < 38 mg/dL and ≥38 mg/dL, and TG < 120 mg/dL and ≥120 mg/dL. The association between TC/HDL-C and all-cause mortality was significant in patients with HDL-C levels < 38 mg/dL (HR, 2.29; 95% CI, 1.13–4.66; *p* = 0.022).

## 4. Discussion

The present study showed an association between TC/HDL-C and all-cause mortality among incident PD patients. Patients with higher TC/HDL-C had worse survival during the follow-up period than other groups, whereas no difference in survival was seen for TC, HDL-C and LDL-C. Subgroup analyses demonstrated higher TC/HDL-C as an independent risk factor for all-cause mortality in patients with older age (≥50 years), females, patients with lower HDL-C levels (<38 mg/dL) and the presence of CVD. A linear regression analysis revealed that patients with elevated TC/HDL-C had higher BMI. Our study suggests that non-traditional lipid profiles such as increased serum TC/HDL-C are independently associated with an increased risk of all-cause mortality in PD patients.

The relationship between TC/HDL-C and cardiovascular events has been investigated in several studies [5,12,13,14,15,16,17,18,19,20,21,22]. The exact pathogenic implication of TC/HDL-C is unknown; however, recent data indicate that TC/HDL-C is correlated with LDL-particle (LDL-P). Particle-based measures such as LDL-P or apolipoprotein B (ApoB) are discordantly greater than LDL-C more frequently in patients with insulin resistance, lower HDL-C, lower LDL-C, higher TG, and those on statins [23,24,25,26]. LDL-P accumulates in the arterial wall and leads to the subsequent subendothelial retention of LDL and other ApoB containing lipoproteins, resulting in atherosclerotic disease. LDL-C levels may insufficiently reflect the process of atheroma formation and underestimate CVD risk. Mathews et al. showed that TC/HDL-C was most correlated with LDL-P in standard lipid profiles [18]. This finding indicates that TC/HDL-C may provide more information about lipoprotein particle concentration and size than the standard lipid profile. Moreover, TC/HDL-C was strongly associated with cardiovascular events in the patients whose ratios of apolipoprotein B_100_ to apolipoprotein A-I, LDL-C, and non-HDL-C levels were discordant [17,19,20,21]. Therefore, our results are consistent with previous studies in that TC/HDL-C was associated with mortality unlike traditional lipid profiles such as TC and HDL-C.

Chronic kidney disease (CKD) causes major alterations in lipid profiles. The distinguishing findings of dyslipidemia in CKD are increased TG, decreased HDL-C, and variable levels of LDL-C. This altered composition of lipoproteins contributes to atherogenicity in patients with CKD [27]. HDL-C levels in both HD and PD patients demonstrate significant impairment of cholesterol efflux, which may be associated with augmented atherosclerosis [28]. However, changes in HDL-C levels may have different impacts on the outcomes of patients on the maintenance of CKD, HD, and PD patients. In a study of CKD patients, TC/HDL-C was independently associated with rapid CKD progression but could not predict mortality [29]. A study of HD patients found that low TC/HDL-C was associated with higher mortality. Increments of HDL-C over 50mg/dL are reportedly paradoxically associated with increased mortality [7]. HDL-C in patients with ESRD on the maintenance of HD is associated with reduced antichemotactic activity and increased macrophage inflammatory cytokine responses, which can increase atheromatosis and CV mortality [30]. Elevated oxidized-HDL-C concentrations are associated with increased CV mortality in HD patients [31].

Contrary to the previous study in HD patients, our results in the present study demonstrated higher TC/HDL-C as a risk factor for all-cause mortality in PD patients, which is consistent with the general population. A previous study also reported that elevated TC/HDL-C could predict survival in 125 patients with continuous ambulatory peritoneal dialysis [32]. These conflicting results may be explained by heterogeneous lipid profiles among HD and PD patients. Reduced lecithin-cholesterol acyltransferase (LCAT) activity may also contribute to delayed HDL-C maturation in patients on HD, leading to decreased cholesterol efflux. However, HDL-C in PD patients is associated with low paraoxonase activity, which is associated with impaired endothelial nitric oxide production and loss of the endothelial anti-inflammatory and endothelial repair-stimulating effects of HDL [31]. The significant reduction in paraoxonase activity can be explained by increased glycation of HDL-C in PD patients [33,34]. Therefore, the composition, function, and enzymatic activities of HDL-C could be altered by dialysis modalities. Although the clinical implication of differences in HDL-C has yet to be fully elucidated, such differences may explain the different impact of HDL-C on the survival of HD and PD patients.

Some registry studies reported that PD has a higher mortality rate than HD among patients with cardiovascular disease [35,36]. Although the SHARP study did not sufficiently confirm the effect of the lipid-lowering agent and was underpowered for PD patients, PD patients showed a much larger effect size than other subgroups [37]. Further studies are needed to confirm the effect of lipid-lowering agents on PD patients for which the mortality could be affected by the combined cardiovascular condition and high TC/HDL-C.

A linear regression analysis showed a positive correlation between TC/HDL-C and BMI. Although the effect of BMI on the survival of PD patients is controversial, several studies reported that obesity is associated with a higher risk of all-cause mortality and CVD mortality [38,39]. An additional subgroup analysis was performed based on BMI 23 kg/m^2^, but no significant results were found (Figure 2). It is conceivable that high TC/HDL-C has no synergistic or additive effect on mortality in patients with high BMI. Statistical significance may not appear because the number of the patients and events in the subgroup according to BMI is not sufficient. However, our study suggests that higher TC/HDL-C could be a risk factor such as BMI for mortality in patients on the maintenance of PD.

However, unlike all-cause mortality, we found no significant association between TC/HDL-C and CV mortality in our study. This may be explained by the relatively low numbers of deaths from CVD in our study, whereas, in the subgroup analysis, CVD patients were predisposed to have higher all-cause mortality rate in the group with a higher TC/HDL-C ratio compared to patients without CVD. Further studies with a larger number of patients are warranted to elucidate the correlation between TC/HDL-C and CV mortality in PD patients.

Further, we found that increased serum TC/HDL-C was an independent risk factor for mortality in the subgroup of old age, female and HDL-C levels less than 38 mg/dL. Old age and female sex are associated with a higher risk of death in patients with PD than with HD [40,41,42]. Although there are few studies about the role of HDL-C in PD patients [9], low HDL-C is a risk factor for mortality in the general population [43,44]. These suggest that the particular subgroups with increased mortality might be more susceptible to altered cholesterol metabolism such as high TC/HDL-C. Further studies are needed to evaluate the impact of TC/HDL-C on mortality in high-risk subgroups.

The strengths of this study are its prospective nature with a relatively long follow-up of up to 6 years and the availability of detailed data regarding baseline characteristics, comorbidities, and laboratory variables. However, there are some limitations to this study. First, given the observational nature of this study design, the results should be interpreted accordingly. Second, we did not examine other atherogenic lipoproteins such as Apo B100, which may be more specific to LDL-P. Third, our study comprised only Korean patients, so the conclusions cannot be generalized to other ethnicities. Fourth, this study lacked several important peritoneal dialysis data values that could affect lipid profiles such as peritoneal solute transfer rates, glucose prescription, use of icodextrin, and glycemic control at baseline. Fifth, the use of statins was not investigated in our baseline data. However, several studies show that statin therapy did not significantly reduce the risk of major atherosclerotic events in dialysis patients [37,45,46]. Thus, the prescription for statins would not have much effect on our study results.

## 5. Conclusions

Higher TC/HDL-C was significantly associated with an increased risk of all-cause mortality in incident PD patients. Further, single lipid markers alone such as TC or HDL-C were unable to predict mortality in PD patients. Although the underlying mechanisms are unclear, the use of inexpensive and readily available non-traditional lipid profiles such as TC/HDL-C may inform risk of mortality in patients with PD.

## Figures and Tables

**Figure 1 nutrients-14-00144-f001:**
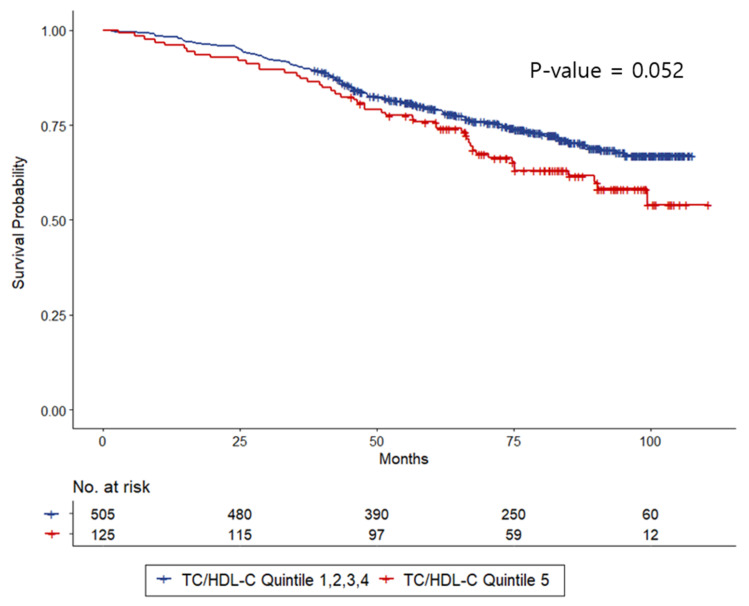
Kaplan–Meier curve of all-cause mortality according to TC/HDL-C.

**Figure 2 nutrients-14-00144-f002:**
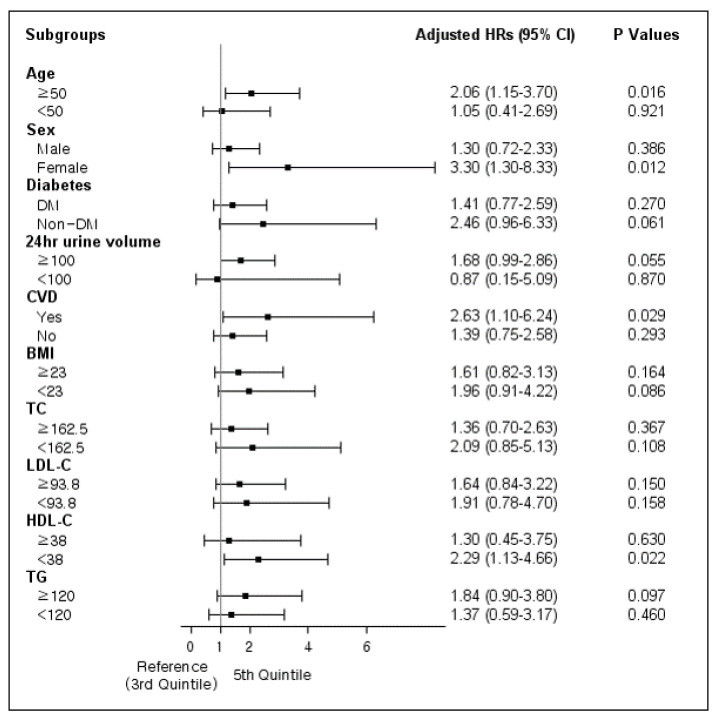
Subgroup analysis according to age, sex, presence of DM, 24 h urine volume, presence of CVD, BMI, and serum lipid levels.

**Table 1 nutrients-14-00144-t001:** Baseline characteristics of individuals by baseline serum TC/HDL-C ratio quintile.

Variables	TC/HDL-C					
	1st	2nd	3rd	4th	5th	*p*-Value
Age at initiation of dialysis (years)	51.61 ± 14.45	52.46 ± 12.19	49.95 ± 11.90	51.05 ± 12.85	52.58 ± 11.99	0.451
Sex (Male, %)	69 (54.8)	74 (59.2)	74 (58.23)	75 (59.1)	82 (65.6)	0.526
Body mass index (kg/m^2^)	21.94 ± 3.24	22.39 ± 2.92	23.13 ± 3.22	23.66 ± 3.40	24.36 ± 3.34	<0.001
Primary renal disease, n (%)						
Diabetes	50 (39.7)	50 (40.0)	45 (35.4)	52 (40.9)	70 (56.0)	0.034
Hypertension	27 (21.4)	27 (21.6)	21 (16.5)	25 (19.7)	16 (12.8)	
Glomerulonephritis	26 (20.6)	30 (24.0)	24 (18.9)	24 (18.9)	19 (15.2)	
Others	23 (18.3)	18 (14.4)	37 (29.1)	26 (20.5)	20 (16.0)	
Comorbidity at initiation of dialysis, n (%)						
Congestive heart failure	9 (7.1)	12 (9.7)	13 (10.2)	14 (11.0)	11 (8.8)	0.858
Coronary artery disease	15 (11.9)	14 (11.2)	7 (5.6)	12 (9.5)	12 (9.6)	0.473
Peripheral vascular disease	7 (5.6)	7 (5.6)	7 (5.7)	5 (3.9)	4 (3.2)	0.838
Arrhythmia	4 (3.2)	0 (0.0)	2 (1.6)	3 (2.4)	1 (0.8)	0.285
Cerebrovascular disease	5 (15.6)	5 (15.6)	8 (6.3)	7 (5.6)	7 (5.6)	0.887
Chronic lung disease	6 (4.8)	2 (1.6)	3 (2.4)	6 (4.7)	3 (2.4)	0.476
Peptic ulcer disease	1 (0.79)	7 (5.6)	10 (7.87)	3 (2.36)	3 (2.4)	0.023
Moderate to severe chronic liver disease	9 (7.1)	6 (4.8)	2 (1.6)	1 (0.8)	3 (2.4)	0.032
Connective tissue disease	9 (7.1)	10 (8.0)	16 (12.6)	14 (11.0)	14 (11.2)	0.558
Malignancy	5 (4.0)	5 (4.0)	3 (2.4)	5 (4.0)	2 (1.6)	0.723
Modified Charlson comorbidity index	4.55 ± 2.26	5.55 ± 2.26	4.28 ± 2.05	4.41 ± 2.13	4.69 ± 1.98	0.607
Laboratory data at initiation of dialysis						
Hemoglobin (g/dL)	9.34 ± 1.87	9.67 ± 1.49	9.76 ± 1.96	9.61 ± 1.45	9.72 ± 1.51	0.282
Blood urea nitrogen (mg/dL)	77.27 ± 34.99	82.58 ± 40.13	74.76 ± 34.29	69.99 ± 31.19	69.64 ± 30.24	0.015
Creatinine (mg/dL)	8.42 ± 3.55	9.06 ± 3.96	8.44 ± 3.29	8.67 ± 3.53	9.30 ± 4.19	0.236
Albumin (g/dL)	3.38 ± 0.58	3.58 ± 0.55	3.52 ± 0.59	3.55 ± 0.49	3.70 ± 0.63	0.016
Calcium (mg/dL)	7.99 ± 0.98	8.02 ± 0.99	7.96 ± 1.00	8.10 ± 1.08	8.17 ± 0.94	0.458
Phosphorus (mg/dL)	5.49 ± 1.93	5.37 ± 1.56	5.14 ± 1.89	5.35 ± 1.70	5.48 ± 1.75	0.517
Urine volume (mL/day)	920.91 ± 575.85	989.49 ± 692.30	1000.31 ± 653.44	937.47 ± 623.06	885.30 ± 647.41	0.586

**Table 2 nutrients-14-00144-t002:** Relationship between serum TC/HDL-C ratio and mortality.

	Quintile 1		Quintile 2		Quintile 4		Quintile 5	
	HR (95% CI)	*p*-Value	HR (95% CI)	*p*-Value	HR (95% CI)	*p*-Value	HR (95% CI)	*p*-Value
Model 1	1.12 (0.77–2.08)	0.347	1.40 (0.87–2.27)	0.180	1.36 (0.84–2.21)	0.208	1.74 (1.09–2.76)	0.020
Model 2	1.17 (0.71–1.95)	0.535	1.22 (0.75–2.00)	0.420	1.30 (0.80–2.11)	0.286	1.65 (1.04–2.64)	0.035
Model 3	1.27 (0.76–2.13)	0.366	1.29 (0.79–2.12)	0.315	1.46 (0.88–2.40)	0.141	1.70 (1.04–2.76)	0.034
Model 4	1.27 (0.75–2.13)	0.374	1.29 (0.78–2.12)	0.319	1.45 (0.88–2.34)	0.148	1.69 (1.04–2.76)	0.036

HR, hazard ratio. Reference group was Quintile 3. Model 1: Unadjusted. Model 2: Model 1 plus age, sex, and body mass index. Model 3: Model 2 plus laboratory data and MCCI. Model 4: Model 3 plus 24 h urine volume.

**Table 3 nutrients-14-00144-t003:** Univariate and multivariate linear regression of TC/HDL-C.

	Univariate					Multivariate				
	β	Std	t	*p*-Value	R^2^	β	Std	t	*p*-Value	R^2^
(intercept)						1.840	0937	1.96	0.0500	0.036
BMI	0.129	0.030	4.38	<.0001	0.030	0.125	0.031	4.09	<0.0001	
DM	0.287	0.200	1.44	0.1510	0.003	0.120	0.215	0.56	0.5762	
PUD	−0.242	0.522	−0.46	0.6432	0.000	−0.143	0.518	−0.28	0.7832	
CLD	0.021	0.571	0.04	0.9702	0.000	0.060	0.567	0.11	0.9158	
BUN	−0.006	0.003	−2.06	0.0397	0.007	−0.005	0.003	−1.75	0.0807	
Albumin	0.012	0.175	0.07	0.9477	0.000	0.041	0.187	0.22	0.8265	

**Table 4 nutrients-14-00144-t004:** Relationship between total serum cholesterol and mortality.

	Quintile 1 (N = 126)		Quintile 2 (N = 125)		Quintile 4 (N = 127)		Quintile 5 (N = 125)	
	HR (95% CI)	*p*-Value	HR (95% CI)	*p*-Value	HR (95% CI)	*p*-Value	HR (95% CI)	*p*-Value
Model 1	1.12 (0.71–1.77)	0.616	0.99 (0.62–1.57)	0.953	0.97 (0.61–1.53)	0.888	1.74 (1.09–2.76)	0.786
Model 2	0.94 (0.59–1.49)	0.793	0.92 (0.58–1.46)	0.722	0.91 (0.57–1.44)	0.683	0.87 (0.55–1.40)	0.570
Model 3	0.76 (0.47–1.24)	0.271	0.90 (0.56–1.45)	0.664	0.79 (0.49–1.26)	0.315	0.73 (0.45–1.17)	0.188
Model 4	0.77 (0.47–1.24)	0.278	0.90 (0.56–1.45)	0.678	0.79 (0.50–1.27)	0.331	0.73 (0.45–1.18)	0.198

HR, hazard ratio. Reference group was Quintile 3. Model 1: Unadjusted. Model 2: Model 1 plus age, sex, and body mass index. Model 3: Model 2 plus laboratory data and MCCI. Model 4: Model 3 plus 24 h urine volume.

**Table 5 nutrients-14-00144-t005:** Relationship serum HDL-C and mortality.

	Quintile 1 (N = 126)		Quintile 2 (N = 125)		Quintile 4 (N = 127)		Quintile 5 (N= 125)	
	HR (95% CI)	*p*-Value	HR (95% CI)	*p*-Value	HR (95% CI)	*p*-Value	HR (95% CI)	*p*-Value
Model 1	1.39 (0.90–2.15)	0.136	0.79 (0.49–1.27)	0.328	1.24 (0.80–1.93)	0.345	0.70 (0.42–1.17)	0.170
Model 2	1.32 (0.85–2.06)	0.214	0.87 (0.53–1.42)	0.586	1.28 (0.82–2.01)	0.273	0.80 (0.47–1.34)	0.392
Model 3	1.23 (0.78–1.94)	0.371	0.81 (0.49–1.33)	0.405	1.17 (0.74–1.85)	0.497	0.70 (0.41–1.18)	0.175
Model 4	1.21 (0.77–1.93)	0.398	0.80 (0.49–1.33)	0.393	1.17 (0.74–1.84)	0.506	0.69 (0.41–1.17)	0.171

HR, hazard ratio. Reference group was Quintile 3. Model 1: Unadjusted. Model 2: Model 1 plus age, sex, and body mass index. Model 3: Model 2 plus laboratory data and MCCI. Model 4: Model 3 plus 24 h urine volume.

## Data Availability

The data of this study are available from the corresponding author on reasonable request.

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
