# Peer review of "Higher Serum Total Cholesterol to High-Density Lipoprotein Cholesterol Ratio Is Associated with Increased Mortality among Incident Peritoneal Dialysis Patients"

_nutrients, 2021, doi:10.3390/nu14010144_

Round 1

Reviewer 1 Report

The authors described the association of the TC/HDL-C ratio in serum with mortality in incident peritoneal dialysis (PD) patients. The highest quintile of TC/HDL-C was associated with increased risk of all-cause mortality, whereas TC and HDL were not associated with mortality. The paper is clearly written and the methodology appears well performed. However the results and their interpretation brings up some points of discussion.

  1. TC consists for a large part of LDL-C. In healthy subjects HDL-C counts for about 20%. In case of atherosclerosis, elevated TC is the result of predominantly elevated LDL-C. A high TC/HDL-C ratio in the fifth quintile is the result of a high TC value due to elevated LDL-C in combination with a low HDL-C value. Although TC increases and HDL-C decreases,  they do not relate to all-cause mortality. However, TC/HDL-C increases much stronger due to the combined effects. That may explain the relationship with all-cause mortality. A question is why this association would be limited to the PD patients. One expects the same for non-dialysis patients. What is known?
  2. The authors used the third quintile as the reference. In this group the TC/LDL-C ratio is intermediate and represents the average PD patient. The authors ignored the first quintile with lowest TC/LDL-C values. This group represents the patients with lowest TC and highest HDL-C and should have the lowest mortality rate. It would strengthen the paper when these data are included.
  3. A linear positive relationship between TC/HDL-C and BMI has been found. The authors describe the TC/HDL-C ratio as an independent marker to predict all-cause mortality. The strong relationship with BMI must mean that obesity is the risk factor for mortality which is expressed by the TC/HDL-C ration. Could BMI be a predictor with similar sensitivity? The discussion about this relationship indicates BMI as the determining factor. “Linear regression analysis showed a positive correlation between TC/HDL-C and BMI. Although the effect of BMI on the survival of PD patients is controversial, several studies have reported obesity is associated with higher risk of all-cause mortality and CVD mortality [34,35]. The results of the present study indicate higher TC/HDL-C, in addition to BMI, may be a risk factor for mortality in patients on maintenance PD.” The last statement “in addition” may be incorrect when the strong relationship exists.
  4. In fig. 2 the increased risks obtained with the highest quintile TC/HDL-C ratio are shown in subgroups. The BMI was not considered a sub group. Two groups should be considered, maybe < 23 and > 23. It may be expected that the BMI group>23 will have a higher risk. The interpretation of the cause of increased risk in a certain subgroup may be discussed. The particular subgroups with increased risk are subgroups with known altered cholesterol metabolism.

Author Response

Manuscript ID: nutrients-1475510

Title: Higher serum total cholesterol to high-density lipoprotein cholesterol ratio is associated with increased mortality among incident peritoneal dialysis patients

Dear Editor:

We received the evaluation of our manuscript by the reviewers and were pleased that they are willing to evaluate a revised version. We appreciate the reviewers for providing us with guidelines to improve the quality of our paper. We have made alterations in the text to address the concerns of the reviewers appropriately. Please find herein detailed answers to all the issues that were raised. The changes that have been made are shown in the revised manuscript file using the “Track Changes” function of Microsoft Word.

Please do not hesitate to contact us if further clarification should be required.

Thank you for evaluating our manuscript and providing us with important comments and suggestions that have contributed to the improvement.

Jang-Hee Cho, MD, PhD

Professor

Division of Nephrology, Department of Internal Medicine, Kyungpook National University Hospital, 130 Dongdeok-ro, Jung-gu, Daegu, 41944, Korea

Tel: +82-53-200-5550

Fax: +82-53-426-2046

Email: jh-cho@knu.ac.kr

Response to Reviewer 1 Comments

The authors described the association of the TC/HDL-C ratio in serum with mortality in incident peritoneal dialysis (PD) patients. The highest quintile of TC/HDL-C was associated with increased risk of all-cause mortality, whereas TC and HDL were not associated with mortality. The paper is clearly written and the methodology appears well performed. However the results and their interpretation brings up some points of discussion.

Point 1: TC consists for a large part of LDL-C. In healthy subjects HDL-C counts for about 20%. In case of atherosclerosis, elevated TC is the result of predominantly elevated LDL-C. A high TC/HDL-C ratio in the fifth quintile is the result of a high TC value due to elevated LDL-C in combination with a low HDL-C value. Although TC increases and HDL-C decreases, they do not relate to all-cause mortality. However, TC/HDL-C increases much stronger due to the combined effects. That may explain the relationship with all-cause mortality. A question is why this association would be limited to the PD patients. One expects the same for non-dialysis patients. What is known?

Response 1: We would like to thank you for taking the time to review our study. As per your recommendation, we searched studies about the non-traditional lipid profiles on non-dialysis patients. Recent studies which showed the correlation between triglyceride/high-density lipoprotein cholesterol (TG/HDL-C) ratio and mortality were found [Kim et al. Predictive value of triglyceride/high-density lipoprotein cholesterol for major clinical outcomes in advanced chronic kidney disease. Clin Kidney J. 2020; 14: 1961. PMID 34345420, Lee et al. Lipid profiles and risk of major adverse cardiovascular events in CKD and diabetes. Plos One. 2020; 15: e0231328. PMID 32271842], however, there was few studies about TC/HDL-C on non-dialysis patients [Ali et al. Predictive factors of rapid linear renal progression and mortality in patients with chronic kidney disease. BMC Nephrol. 2020; 21: 345. PMID 32795261]. In this study, higher TC/HDL-C was independently associated with rapid CKD progression, but not with mortality [PMID 32795261]. Therefore, the influences of TC/HDL-C on non-dialysis CKD patients are not fully known, and additional research would be needed. As you pointed out, we have additionally described the study about the TC/HDL-C in the non-dialysis patients at the ‘Discussion’ section, page 8, line 192 as follows: 

“In a study of CKD patients, higher TC/HDL-C was an independent risk factor for rapid CKD progression but could not predict mortality.”

Point 2: The authors used the third quintile as the reference. In this group the TC/LDL-C ratio is intermediate and represents the average PD patient. The authors ignored the first quintile with lowest TC/LDL-C values. This group represents the patients with lowest TC and highest HDL-C and should have the lowest mortality rate. It would strengthen the paper when these data are included.

Response 2:

Thank you for your helpful advice. When the first group was used as a reference, as in your comments, there was no statistically significant difference from other groups, and the third group showed the lowest HR. These results are consistent with the analysis using the third quintile group as a reference. Considering these facts, TC/HDL-C might be meaningful as a risk factor only when it shows a very high value. To help readers understand, the results have been added at the ‘Results’ section, page 4, line 128 as follows and as the supplement Table 1:

"When the quintile 1 was considered as a reference group, there was no statistically significant difference from other groups (supple Table 1).”

Supplement Table 1. Relationship between serum TC/HDL-C ratio and mortality

Quintile 2

Quintile 3

Quintile 4

Quintile 5

HR (95%CI)

p-value

HR (95% CI)

p-value

HR(95% CI)

p-value

HR(95%CI)

p-value

Model 1

1.10 (0.69-1.75)

0.689

0.79 (0.48-1.29)

0.347

1.08 (0.68-1.71)

0.759

1.37 (0.88-2.13)

0.163

Model 2

1.04 (0.65-1.67)

0.862

0.85 (0.51-1.42)

0.535

1.11 (0.69-1.79)

0.670

1.41 (0.88-2.25)

0.153

Model 3

1.02 (0.63-1.64)

0.948

0.79 (0.47-1.32)

0.366

1.15 (0.71-1.86)

0.579

1.34 (0.83-2.15)

0.232

Model 4

1.02 (0.63-1.64)

0.943

0.79 (0.47-1.33)

0.374

1.15 (0.71-1.86)

0.582

1.34 (0.83-2.15)

0.233

HR, hazard ratio. Reference group was Quintile 1. Model 1: Unadjusted. Model 2: Model 1 plus age, sex and body mass index. Model 3: Model 2 plus laboratory data and MCCI.  Model 4: Model 3 plus 24 hour urine volume.

Point 3:  A linear positive relationship between TC/HDL-C and BMI has been found. The authors describe the TC/HDL-C ratio as an independent marker to predict all-cause mortality. The strong relationship with BMI must mean that obesity is the risk factor for mortality which is expressed by the TC/HDL-C ration. Could BMI be a predictor with similar sensitivity? The discussion about this relationship indicates BMI as the determining factor. “Linear regression analysis showed a positive correlation between TC/HDL-C and BMI. Although the effect of BMI on the survival of PD patients is controversial, several studies have reported obesity is associated with higher risk of all-cause mortality and CVD mortality [34,35]. The results of the present study indicate higher TC/HDL-C, in addition to BMI, may be a risk factor for mortality in patients on maintenance PD.” The last statement “in addition” may be incorrect when the strong relationship exists.

Response 3: Thank you for your comment. As you pointed out, we checked the correlation between TC/HDL-C and mortality, not BMI and mortality. So, the sentence has been corrected as follows to avoid misunderstanding at ‘Discussion’ section, page 8, line 226 as follows:

“Our study suggests that higher TC/HDL-C could be a risk factor such as BMI for mortality in patients on the maintenance PD.”

Point 4:  In fig. 2 the increased risks obtained with the highest quintile TC/HDL-C ratio are shown in subgroups. The BMI was not considered a sub group. Two groups should be considered, maybe < 23 and > 23. It may be expected that the BMI group>23 will have a higher risk. The interpretation of the cause of increased risk in a certain subgroup may be discussed. The particular subgroups with increased risk are subgroups with known altered cholesterol metabolism.

Response 4: Thank you for your detailed comment. As per your recommendation, we added BMI to the subgroup analyses, but there was no difference according to the BMI category. The results have been applied to Figure 2.

We have additionally described the cause of increased risk in a certain subgroup at ‘Discussion’ section, page 9, line 237 as follows:

“Further, we found that increased serum TC/HDL-C was an independent risk factor for mortality in the subgroup of old age, female and HDL-C levels less than 38mg/dL. Old age and female sex are associated with a higher risk of death in patients with PD than with HD [PMID 20876398, 22391139, 25941194]. Although there are few studies about the role of HDL-C in PD patients [PMID  28669685], low HDL-C is a risk factor for mortality in the general population [PMID 32283957, 33398433]. These suggest that the particular subgroups with increased mortality might be more susceptible to altered cholesterol metabolism such as high TC/HDL-C. Further studies are needed to evaluate the impact of TC/HDL-C on mortality in high-risk subgroups.“

Thank you for providing us with suggestions to improve the quality of our paper. We hope that this revised version will be met with your approval.

Reviewer 2 Report

This is a fairly straight forward cohort study looking at the association between TC:HDL ratio in incident Korean patients to subsequent survival. The finding is not in fact entirely novel (see: Little J, Phillips L, Russell L, Griffiths A, Russell GI, Davies SJ. Longitudinal lipid profiles on CAPD: their relationship to weight gain, comorbidity, and dialysis factors. J Am Soc Nephrol. 1998 Oct;9(10):1931-9. doi: 10.1681/ASN.V9101931.  ) but does confirm this relationship in a larger PD population of different ethnicity - and as the authors point out, the fact that this is different to HD patients is of interest. There is a suspicion that PD patients have more lipid-related cardiovascular risk (e.g. ANZDATA studies of this) - and the main value of this study, I believe, is in pointing out the need for further trials of lipid lowering agents in PD patients (it is worth mentioning that the SHARP trial was underpowered for PD patients, but actually had a much larger effect size than other subgroups). I think you could make more of this in the discussion...

I have some questions and would point out some limitations.

I am not clear why the initial plan was to divide the analysis into quintiles. Was this a pre-defined approach? - and if so why? I suspect it was made after exploring the data - but if so, that process should be explained in the methods. TC:HDL is a continuous variable - so the final cut-off value (i.e. 5.6) may in fact not be optimal one - and understanding of the relationship between the ratio and outcomes (i.e. linear/nonlinear would be useful), e.g. for the design of future trials.

It is not clear from the text whether any of the patients were on statins - it would be surprising if not (for secondary prevention). If there is a clear policy to stop statins on starting PD? - i.e. before these samples were measured, that would be important to know.

The survival models do not include peritoneal solute transfer rates at baseline - and we should remember that a number of other potentially important covariates are missing that might affect lipid profiles...e.g. glucose prescription, use of icodextrin, glycemic control - and that these change longitudinally over time (as does weight/BMI - see Little et al.) These issues should be acknowledged under limitations.

A couple of minor points: Line 24. in the abstract "However, non-traditional lipid profiles, such as an increased serum TC/HDL-C" suggests that a number of other non-traditional profiles were examined, but only TC/HDL was investigates.

Line 120: "Multivariate Cox regression analysis found increasing patient survival with increasing TC/HDL-C in PD patients (Table 2)."  I think the should be decreasing survival.

Author Response

Manuscript ID: nutrients-1475510

Title: Higher serum total cholesterol to high-density lipoprotein cholesterol ratio is associated with increased mortality among incident peritoneal dialysis patients

Dear Editor:

We received the evaluation of our manuscript by the reviewers and were pleased that they are willing to evaluate a revised version. We appreciate the reviewers for providing us with guidelines to improve the quality of our paper. We have made alterations in the text to address the concerns of the reviewers appropriately. Please find herein detailed answers to all the issues that were raised. The changes that have been made are shown in the revised manuscript file using the “Track Changes” function of Microsoft Word.

Please do not hesitate to contact us if further clarification should be required.

Thank you for evaluating our manuscript and providing us with important comments and suggestions that have contributed to the improvement.

Jang-Hee Cho, MD, PhD

Professor

Division of Nephrology, Department of Internal Medicine, Kyungpook National University Hospital, 130 Dongdeok-ro, Jung-gu, Daegu, 41944, Korea

Tel: +82-53-200-5550

Fax: +82-53-426-2046

Email: jh-cho@knu.ac.kr

Response to Reviewer 2 Comments

This is a fairly straight forward cohort study looking at the association between TC:HDL ratio in incident Korean patients to subsequent survival. The finding is not in fact entirely novel (see: Little J, Phillips L, Russell L, Griffiths A, Russell GI, Davies SJ. Longitudinal lipid profiles on CAPD: their relationship to weight gain, comorbidity, and dialysis factors. J Am Soc Nephrol. 1998 Oct;9(10):1931-9. doi: 10.1681/ASN.V9101931.  ) but does confirm this relationship in a larger PD population of different ethnicity - and as the authors point out, the fact that this is different to HD patients is of interest. There is a suspicion that PD patients have more lipid-related cardiovascular risk (e.g. ANZDATA studies of this) - and the main value of this study, I believe, is in pointing out the need for further trials of lipid lowering agents in PD patients (it is worth mentioning that the SHARP trial was underpowered for PD patients, but actually had a much larger effect size than other subgroups). I think you could make more of this in the discussion...

Response: Thank you for taking the time to review our study and providing good references. As per your recommendation to improve the “Discussion” section's clarity, we have cited and summarized relevant references.

As you pointed out, our finding is not entirely novel. We have described the previous study to evaluate the relationship between TC/HDL-C and mortality in CAPD patients at the ‘Discussion’ section, page 8, line 203 as follows:

“A previous study also reported that elevated TC/HDL-C could predict survival in 125 patients with continuous ambulatory peritoneal dialysis [PMID 9773795]”

We have additionally described the feature of PD patients compared to HD and the need for the further study with lipid-lowering agents at the ‘Discussion’ section, page 8, line 216 as follows:

“Some registry studies reported that PD have higher mortality rate than HD among patients with cardiovascular disease [PMID: 21775972, PMID: 19092128]. Although the SHARP study did not sufficiently confirm the effect of the lipid-lowering agent and was underpowered for PD patients, PD patients showed a much larger effect size than other subgroups [PMID: 21663949]. Further studies are needed to confirm the effect of lipid lowering agents in PD patients of which the mortality could be affected by cardiovascular comorbidity and high TC/HDL-C.”

Point 1: I have some questions and would point out some limitations.

I am not clear why the initial plan was to divide the analysis into quintiles. Was this a pre-defined approach? - and if so why? I suspect it was made after exploring the data - but if so, that process should be explained in the methods. TC:HDL is a continuous variable - so the final cut-off value (i.e. 5.6) may in fact not be optimal one - and understanding of the relationship between the ratio and outcomes (i.e. linear/nonlinear would be useful), e.g. for the design of future trials.

Response 1: Your comments will be of great help to future research to investigate the effect of TC/HDL-C in dialysis patients. As you expect, our preliminary analysis showed that TC/HDL-C does not have a linear relationship with mortality. Rather, TC/HDL-C showed a pattern close to the U-shape relationship with patient death and we divided our data into 5 groups in consideration of the features. As shown in our results, the 3rd quintile showed the lowest HR, and HR increased toward both ends. Since it did not show a significant increase in the first quintile, we thought that TC/HDL-C has clinical significance only when TC/HDL-C is very high.

We have described the relationship between TC/HDL-C and mortality was non-linear and had a U-shape relationship at the 'Materials and Methods’ section, page 2, line 85 as follows.

“Preliminary analysis revealed that TC/HDL-C had a U-shape relationship with mortality rather than a linear relationship.”

Point 2: It is not clear from the text whether any of the patients were on statins - it would be surprising if not (for secondary prevention). If there is a clear policy to stop statins on starting PD? - i.e. before these samples were measured, that would be important to know.

Response 2: Thank you for your comment and we totally agree with you. Unfortunately, the use of statins was not investigated in our cohort data. However, several studies, such as the 4D and AURORA studies, have shown that taking statins had no effect on mortality in hemodialysis patients [PMID 16034009, 19332456]. Additionally, in the SHARP study, statin plus ezetimibe therapy did not significantly reduce the risk of major atherosclerotic events in HD and PD patients [PMID 21095263]. So, the effect of statins is thought to be limited in dialysis patients including our study. Instead, we analyzed the correlation between mortality and LDL-C, which is lowered by statins. As a result, there was no association between LDL-C and mortality as shown in supplement Table 2.

Supplement Table 2. Relationship between serum LDL-C and mortality

Quintile 1

Quintile 2

Quintile 4

Quintile 5

HR (95% CI)

p-value

HR (95% CI)

p-value

HR (95% CI)

p-value

HR (95% CI)

p-value

Model 1

1.38 (0.86-2.20)

0.182

1.24 (0.77-2.00)

0.369

1.10 (0.68-1.77)

0.708

1.34 (0.85-2.12)

0.206

Model 2

1.26 (0.79-2.03)

0.331

1.32 (0.82-2.12)

0.262

1.15 (0.71-1.86)

0.570

1.28 (0.81-2.04)

0.287

Model 3

1.05 (0.65-1.70)

0.847

1.27 (0.78-2.05)

0.338

1.13 (0.69-1.83)

0.628

1.02 (0.63-1.65)

0.950

Model 4

1.05 (0.64-1.70)

0.856

1.26 (0.77-2.05)

0.357

1.12 (0.69-1.83)

0.644

1.02 (0.63-1.65)

0.950

HR, hazard ratio. Reference group was Quintile 3. Model 1: Unadjusted. Model 2: Model 1 plus age, sex and body mass index. Model 3: Model 2 plus laboratory data and MCCI. Model 4: Model 3 plus 24 hour urine volume.

In accordance with the KDIGO 2013 lipid guidelines, most dialysis centers in our study did not initiate or discontinue statins or statin/ezetimibe therapy when starting PD. Therefore, we presumed that the prescription for statins would not have changed much and would not have a significant effect on the study results. Nevertheless, we have added the limitation to the point that the use of statins was not investigated at 'Discussion’ section, page 9, line 252 as follows:

“Fifth, the use of statins was not investigated in our baseline data. However, several studies have shown that statin therapy did not significantly reduce the risk of major atherosclerotic events in dialysis patients [PMID 16034009, 19332456, 21095263]. Thus, the prescription for statins would not have much effect on our study results.”

Point 3: The survival models do not include peritoneal solute transfer rates at baseline - and we should remember that a number of other potentially important covariates are missing that might affect lipid profiles...e.g. glucose prescription, use of icodextrin, glycemic control - and that these change longitudinally over time (as does weight/BMI - see Little et al.) These issues should be acknowledged under limitations.

Response 3: Thank you for your helpful advice. As you pointed out, this study lacked data on peritoneal dialysis which could affect lipid profiles. Therefore, the limitation has been added at ‘Discussion’ section, page 8, line 252 as follows:

Fourth, this study lacked several important peritoneal dialysis data that could affect lipid profiles such as peritoneal solute transfer rates, glucose prescription, use of icodextrin, and glycemic control at baseline.”

Point 4: A couple of minor points: Line 24. in the abstract "However, non-traditional lipid profiles, such as an increased serum TC/HDL-C" suggests that a number of other non-traditional profiles were examined, but only TC/HDL was investigates.

Response 4: Thank you for your advice and we agree with your comment. As pointed out, the sentence has been modified at ‘Abstract’ section, page 1, line 24 as follows:

“However, increased serum TC/HDL-C was independently associated with all-cause mortality in PD patients.”

Point 5:  Line 120: "Multivariate Cox regression analysis found increasing patient survival with increasing TC/HDL-C in PD patients (Table 2)."  I think the should be decreasing survival.

Response 5: We appreciate your comments. As per your recommendation, we have changed the sentence at ‘Result’ section, page 4, line 122 as follows:

“Multivariate Cox regression analysis found decreasing patient survival with increasing TC/HDL-C in PD patients (Table 2).”

Thank you for providing us with suggestions to improve the quality of our paper. We hope that this revised version will be met with your approval.

Round 2

Reviewer 1 Report

I thank the authors for responding to my comments. However, the answers do not improve the quality of the paper.  

Reviewer point 2: The authors used the third quintile as the reference. In this group the TC/LDL-C ratio is intermediate and represents the average PD patient. The authors ignored the first quintile with lowest TC/LDL-C values. This group represents the patients with lowest TC and highest HDL-C and should have the lowest mortality rate. It would strengthen the paper when these data are included.

 Authors response 2:

Thank you for your helpful advice. When the first group was used as a reference, as in your comments, there was no statistically significant difference from other groups, and the third group showed the lowest HR. These results are consistent with the analysis using the third quintile group as a reference. Considering these facts, TC/HDL-C might be meaningful as a risk factor only when it shows a very high value. To help readers understand, the results have been added at the ‘Results’ section, page 4, line 128 as follows and as the supplement Table 1:

"When the quintile 1 was considered as a reference group, there was no statistically significant difference from other groups (supple Table 1).”

Reviewer comment to authors response 2: My intention was that the authors show that the subjects in the first group with low TC/HDL-C have less mortality risk than the subjects in group 3 with median TC/HDL-C. The authors answer brings up more questions. Subjects in the TC/HDL-C levels 1 to 4 have similar HRs. Thus only in group 5, HR is elevated. The higher risk in group 5 is documented when the 3rd group is used as reference since this group has the lowest HR. Is the result the same when group 1 is chosen as the reference? The authors should clarify this point.

Reviewer comment point 4:  In fig. 2 the increased risks obtained with the highest quintile TC/HDL-C ratio are shown in subgroups. The BMI was not considered a sub group. Two groups should be considered, maybe < 23 and > 23. It may be expected that the BMI group>23 will have a higher risk. The interpretation of the cause of increased risk in a certain subgroup may be discussed. The particular subgroups with increased risk are subgroups with known altered cholesterol metabolism.

Authors response 4: Thank you for your detailed comment. As per your recommendation, we added BMI to the subgroup analyses, but there was no difference according to the BMI category. The results have been applied to Figure 2.

 We have additionally described the cause of increased risk in a certain subgroup at ‘Discussion’ section, page 9, line 237 as follows:

“Further, we found that increased serum TC/HDL-C was an independent risk factor for mortality in the subgroup of old age, female and HDL-C levels less than 38mg/dL. Old age and female sex are associated with a higher risk of death in patients with PD than with HD [PMID 20876398, 22391139, 25941194]. Although there are few studies about the role of HDL-C in PD patients [PMID  28669685], low HDL-C is a risk factor for mortality in the general population [PMID 32283957, 33398433]. These suggest that the particular subgroups with increased mortality might be more susceptible to altered cholesterol metabolism such as high TC/HDL-C. Further studies are needed to evaluate the impact of TC/HDL-C on mortality in high-risk subgroups.“

Reviewer comment to authors response 4: It is an interesting result that TC/HDL-C and BMI are highly correlated but that BMI does not show increased risk in group 5 with highest TC/HDL-C. Since BMI is expected to increase mortality risk, this finding is important. Thus, the risk of high BMI and the risk of high TC/HDL-C are not coinciding and apparently independent of the relationship. The authors can discuss this.
